# Tailored Synthesis of Heterogenous 2D TMDs and Their Spectroscopic Characterization

**DOI:** 10.3390/nano14030248

**Published:** 2024-01-23

**Authors:** Jungtae Nam, Gil Yong Lee, Dong Yun Lee, Dongchul Sung, Suklyun Hong, A-Rang Jang, Keun Soo Kim

**Affiliations:** 1Department of Physics and Graphene Research Institute, Sejong University, Seoul 05006, Republic of Korea; goodnjt@sejong.ac.kr (J.N.); rlfdyd2000@naver.com (G.Y.L.); geovi012@gmail.com (D.Y.L.); hong@sejong.ac.kr (S.H.); 2Division of Electrical, Electronic and Control Engineering, Kongju National University, Cheonan 31080, Republic of Korea

**Keywords:** heterostructure, direct synthesis, transition metal dichalcogenide, MoS_2_, WS_2_, spectroscopy

## Abstract

Two-dimensional (2D) vertical van der Waals heterostructures (vdWHs) show great potential across various applications. However, synthesizing large-scale structures poses challenges owing to the intricate growth parameters, forming unexpected hybrid film structures. Thus, precision in synthesis and thorough structural analysis are essential aspects. In this study, we successfully synthesized large-scale structured 2D transition metal dichalcogenides (TMDs) via chemical vapor deposition using metal oxide (WO_3_ and MoO_3_) thin films and a diluted H_2_S precursor, individual MoS_2_, WS_2_ films and various MoS_2_/WS_2_ hybrid films (Type I: Mo_x_W_1−x_S_2_ alloy; Type II: MoS_2_/WS_2_ vdWH; Type III: MoS_2_ dots/WS_2_). Structural analyses, including optical microscopy, Raman spectroscopy, transmission electron microscopy (TEM) with energy-dispersive X-ray spectroscopy, and cross-sectional imaging revealed that the A_1g_ and E_2g_ modes of WS_2_ and MoS_2_ were sensitive to structural variations, enabling hybrid structure differentiation. Type II showed minimal changes in the MoS_2′_s A_1g_ mode, while Types I and III exhibited a ~2.8 cm^−1^ blue shift. Furthermore, the A_1g_ mode of WS_2_ in Type I displayed a 1.4 cm^−1^ red shift. These variations agreed with the TEM-observed microstructural features, demonstrating strain effects on the MoS_2_–WS_2_ interfaces. Our study provides insights into the structural features of diverse hybrid TMD materials, facilitating their differentiation through Raman spectroscopy.

## 1. Introduction

Transition metal dichalcogenides (TMDs), akin to two-dimensional (2D) layered structures like graphene, have attracted considerable attention and have been widely applied—from insulators and semiconductors to superconductors—owing to their diverse properties [1,2,3]. Recent studies have highlighted the application scope of heterostructures formed by vertically or horizontally stacking different types of single TMD materials, showcasing their potential in the development of optoelectronic systems, solar cells, and nanoelectronic devices [4,5,6,7,8,9,10,11]. Vertical van der Waals heterostructures (vdWHs), e.g., MoS_2_/WS_2_ structures, are particularly noteworthy. Engineered through material selection and junction design, vdWHs operate through phenomena such as light-induced separation or recombination of electron–hole pairs, driving active research in photodetection, photocatalysis, and development of solar cells [5,7,8]. While mechanical exfoliation and transfer processes can be employed for vdWH synthesis and characterization [12,13], films obtained through mechanical exfoliation exhibit low yields and have inconsistent thicknesses and nonuniform size distribution, hindering the use of such methods in large-scale production. Additionally, interlayer interfaces are susceptible to contamination during transfer. In contrast, chemical vapor deposition (CVD) provides a facile route for vdWH synthesis by controlling variables such as precursors, temperature, and gas atmosphere. CVD allows easy thickness modulation, scalability, and clean interface formation during synthesis, mitigating contamination issues. As a result, CVD has become the primary method for synthesizing TMD-based vdWHs [14,15,16,17]. Various vertical heterostructures, such as WS_2_/MoS_2_, directly synthesized via CVD, have been reported [18,19,20].

One of the challenges in the synthesis of vdWHs is the limited feedback for establishing optimal synthesis conditions post-microstructure examination. During CVD-based synthesis, variations in temperature, atmosphere, and precursor supply may result in the formation of not only 2D TMD films but also TMD hybridized structures with diverse configurations, resembling alloys or quantum dots, rather than vdWHs. Surface observations through techniques such as optical microscopy (OM), scanning electron microscopy (SEM), and atomic force microscopy (AFM) can discern morphological differences; however, distinguishing between alloyed structures and vdWHs is challenging. Although XPS provides insights into elemental composition and chemical bonding, the differentiation of the structural differences in vertically stacked samples remains arduous. Hence, the precise analysis of fine structures can be achieved by directly observing sample interfaces through transmission electron microscopy (TEM). Given that properties vary with structures, clear distinctions in the detailed structures of CVD-synthesized TMD heterostructures are crucial for optimizing vdWH synthesis conditions. However, conducting TEM analysis for every sample is inefficient.

In this study, we propose a more general analytical method by comparing and analyzing Raman spectra with peaks belonging to the A_1g_ and E_2g_ modes vibrating in vertical and horizontal directions with TEM images to identify the microstructures. Two-dimensional films, such as TMDs, are sensitive to strain at interfaces, leading to the transition of phonon vibration modes at the interface. When materials such as MoS_2_ and WS_2_ exhibit different microstructures in alloys and vdWHs, the strain distribution at the interface may differ. We designed and synthesized various TMD heterostructures, performing Raman spectroscopy to analyze the behavior of phonons under strain in the A_1g_ and E_2g_ modes at the interface, allowing the clear distinction of microstructural differences. Furthermore, we confirmed the identification of microstructures by analyzing the phonon behavior arising from strain differences at the interface of TMD heterostructures synthesized via CVD on SiO_2_/Si substrates. For validation, we synthesized various MoS_2_/WS_2_ hybrid films (Type I: Mo_x_W_1−x_S_2_ alloy; Type II: MoS_2_/WS_2_ vdWH; Type III: MoS_2_ dots/WS_2_) via CVD. The fine microstructures of the synthesized TMD films’ heterostructures were examined by preparing TEM specimens through a focused ion beam. Raman mapping of these microstructures was also performed. By analyzing variations in the mobility of the A_1g_ and E_2g_ modes in relation to differences in microstructures, we were able to establish correlations between the observed changes in phonon behavior and specific microstructural characteristics.

## 2. Materials and Methods

### 2.1. Preparation of TMO Thin Films (WO_3_, MoO_3_) on A SiO_2_ Substrate

A 100 mm-sized 300 nm SiO_2_/Si substrate was cleaned with acetone and isopropyl alcohol in an ultrasonic cleaner for 1 h. Subsequently, the cleaned substrate was positioned on the rotation stage within a thermal evaporator system. Once the base pressure of the chamber dropped below 1 × 10^−6^ Torr, the transition metal oxide (TMO) was evaporated by thermal heating. The TMO was deposited on a SiO_2_ substrate at a rate of 0.1 Å/s and at room temperature.

### 2.2. Synthesis of TMD (MoS_2_, WS_2_) Thin Films under H_2_S–Ar Mixed Gas

Crystalline TMD (MoS_2_, WS_2_) thin films were grown in a CVD system. The thermally deposited TMO (MoO_3_, WO_3_) film on the SiO_2_ substrate was placed at the center of a quartz tube in a furnace. Then, the pressure of the quartz tube was decreased to less than 1 × 10^−3^ Torr, and the quartz tube was purged with Ar (200 sccm). Next, the manual valve was shut until the pressure reached 350 Torr, and the furnace was then heated to 800 °C at a rate of 25 °C min^−1^. Upon reaching 800 °C, the furnace was purged for 10 min with H_2_S gas (0.6 Torr). Subsequently, the chamber was left to naturally cool to room temperature.

### 2.3. Transfer of Hetero-TMD Thin Film onto the TEM Grid

The TMD samples synthesized on the Si substrate were immersed in a 2 mol NaOH aqueous solution for 30 s on a hot plate heated to 80 °C. Upon placing the TMD sample diagonally into deionized water, the TMD thin film separates from the substrate and remains on the water surface. The floating TMD thin film was then carefully transferred onto a TEM grid, left to dry naturally in a desiccator, and was further dried in an oven at 100 °C for 3 h.

### 2.4. Characterization

To investigate the microstructure of the hetero-TMD thin films, TEM was conducted using a spherical aberration-corrected transmission electron microscope (Thermo Fisher Scientific, Waltham, MA, USA, Titan G2 60–300) equipped with a monochromator at an accelerating voltage of 80 kV. Measurements were performed at this acceleration voltage to minimize any knock-on damage and contamination caused by the beam. X-ray photoelectron spectroscopy (XPS) analyses were performed using a K-Alpha XPS system and a monochromatic Al Ka X-ray radiation source (Thermo Fisher Scientific). The elements composing the TMO and TMD thin films (W4f, Mo3d, S2p, C1s, and O1s) were measured by irradiating X-rays with a pass energy of 50 eV in an area of 400 μm^2^_._ Raman spectroscopy (Renishaw, Wotton-under-Edge, Glos, UK, Renishaw Invia Reflex) was performed using a 514 nm laser with an 1800/mm grating to observe the phonon modes of the TMD thin films (WS_2_ and MoS_2_). Recent studies have revealed that the E_2g_ and A_1g_ peaks in the Raman spectrum of MoS_2_ are influenced by mechanical strain and changes in charge carrier concentration, resulting in shifts in their frequencies [13,21]. To examine this, we performed a correlation analysis between the E_2g_ and A_1g_ peaks of the MoS_2_ thin films to study the influence of the hetero-TMD structure on the mechanical deformation of the samples. For this analysis, Raman mapping was conducted on a 60 μm × 60 μm area at 1 μm intervals. Additionally, deconvolution of the E_2g_ and A_1g_ peaks in the Raman spectrum of the hetero-TMD thin films, specifically WS_2_ and MoS_2_, was conducted to extract their respective frequencies.

## 3. Results and Discussion

Figure 1 shows the synthesis and characteristics of the three samples. The metal oxide thin films (MoO_3_ or WO_3_, Sigma Aldrich, St. Louis, MI, USA) were deposited on SiO_2_ substrates using a thermal evaporator. During the deposition of the MoO_3_ film, efforts were made to create a patterned structure on the SiO_2_ substrate. A TEM grid served as a shadow mask to create a honeycomb-shaped pattern, ensuring precise positioning for comparative analyses. The use of metal oxides instead of metals is advantageous owing to their lower melting points, enabling reduced processing temperatures. The TMD samples used for analysis were prepared by loading the MoO_3_ and WO_3_ films into a CVD chamber, lowering the pressure using a vacuum pump, and supplying diluted H_2_S gas (2% H_2_S diluted in Ar) at 800 °C under 300 Torr pressure with argon for 10 min. This method was used to synthesize the combination depicted in Figure 1. The number of TMD layers is affected by the thickness of the metal oxide; by adjusting the thickness (number of layers) of TMD in proportion to the thickness of TMO, we sulfurized TMO (MoO_3_, WO_3_) of 1, 3, and 5 nm thicknesses to produce the corresponding TMD (MoS_2_, WS_2_). After synthesis, the Raman spectrum was confirmed (Appendix A and Appendix A). The positions and ratios of the E_2g_ and A_1g_ peaks show that the number of TMD layers increases with the oxide thickness. The results from Appendix A indicate the feasibility of synthesizing TMD from TMO of 1 nm thickness. However, as shown in Appendix A, the TEM observations reveal that WS_2_ films synthesized from 1 nm WO_3_ exhibit a non-uniform structure with small grain sizes and numerous holes. In contrast, WS_2_ films synthesized using a 3 nm-thick WO_3_ precursor exhibit a relatively uniform film shape. Furthermore, examination of the folded regions in the WS_2_ film indicated the presence of four layers. Therefore, in the synthesis of heterogenous TMD, the thickness of the etch metal oxide was set at 3 nm for conducting the experiments.

First, for Mo_x_W_1−x_S_2_ (Type I), we deposited WO_3_ and MoO_3_ films sequentially, each with a thickness of 3 nm, onto the substrate using a thermal evaporator. Following this, they were loaded into a chamber and synthesized at 800 °C using dilute H_2_S gas for sulfurization. Second, the fabrication of MoS_2_/WS_2_ vdWH (Type II) involved a two-step synthesis. Initially, we sulfurized only the WO_3_ film (3 nm) to form the WS_2_ film. Then, we deposited MoO_3_ (3 nm) onto the WS_2_ film, which was subsequently sulfurized. Finally, for the fabrication of MoS_2_ dots/WS_2_ (Type III), after synthesizing the WS_2_ film, the quartz substrate (thickness = 2 mm) was positioned in an inverted orientation. A second layer of transition metal oxide, MoO_3_ film, was placed using a quartz substrate as a spacer. The MoO_3_, which vaporized at 800 °C under a low pressure of 600 mTorr, reacted with the H_2_S on the WS_2_ surface, resulting in its conversion into droplet-shaped MoS_2_.

Figure 2 shows the optical images of the Type I, II and III samples, along with the mapping results of the intensity of the E_2g_ Raman peak signals for MoS_2_ and WS_2_. Noticeable shading differences are evident in the OM images (Figure 2a) for each sample. To analyze these differences, Raman mapping was conducted at a resolution of 1 µm over a 60 μm × 60 µm area centered around the edges of the samples patterned in a hexagonal grid structure. The mapping results revealed clear boundaries between the MoS_2_ and WS_2_, indicating no horizontal growth during the sulfurization of the MoO_3_ film. Furthermore, the Type I, II and III samples exhibited both E_2g_ and A_1g_ peaks for the MoS_2_ and WS_2_, enabling easy differentiation of the MoS_2_ and WS_2_ films [20]. Nevertheless, variations in the peak positions and signal intensities of the E_2g_ and A_1g_ modes were observed in the Raman spectra, as depicted in Figure 2b, depending on the synthesis method. The differences in color observed in the OM images and Raman spectra among the Type I, II and III samples signify distinct morphological or structural states. 

XPS analysis was performed to investigate the chemical composition and binding energy of the MoS_2_ and WS_2_ thin films synthesized from metal oxides along with Type I, II and III samples. All the XPS spectra were based on the C1s peak (284.5 eV) to correct for charging effects. The XPS spectral changes before and after TMD synthesis of the metal oxide thin films are shown in Appendix A. In the W4f spectrum of the WO_3_ thin film, the doublet peaks of 37.6 and 35.8 eV correspond to W4f_5/2_ and W4f_7/2_, respectively, and indicate the W^6+^ of WO_3_. After sulfidation into a WS_2_ thin film, the W^6+^ phase decreases in the W4f spectrum, generating a W^4+^ doublet peak corresponding to the hexagonal (2H) phase of WS_2_ located at 34.6 and 32.5 eV [22]. In the MoO_3_ thin film (Mo3d spectrum), the Mo^5+^ phase located at 235.1 and 231.9 eV, and the Mo^6+^ phase located at 235.5 and 232.4 eV, corresponding to the oxide state, were confirmed. However, after sulfurization, the oxide phase decreased and appeared together with the Mo^4+^ peak, whereas the S2s peak was located at 226.8 eV, indicating the formation of MoS_2_. The Mo^4+^ phase was divided into a tetragonal (1T) phase corresponding to 232.6 and 229.7 eV and a 2H phase corresponding to 233.1 and 229.8 eV through peak deconvolution [23,24,25]. The MoS_2_ thin film was analyzed in a form in which the 1T and 2H phases coexist. And S2p peak was located at 163.25 and 162.1 eV. The ratio of transition metal to sulfur in WS_2_ and MoS_2_ films showed an approximate ratio of 1:2.

Figure 3 shows the XPS spectra of the MoS_2_/WS_2_ hybrid thin films: (a) W4f, (b) Mo3d, and (c) S2p of Type I, II and III samples. In Figure 3a, unlike in the WS_2_ thin film, doublet peaks at 38.6 and 36.8 eV are added in the W4f spectrum of Types I, II and III, corresponding to the peaks of Mo4p [26]. Except for the W4f XPS spectrum, no significant changes were observed in the Mo3d and S2p XPS spectra. The binding energy and elemental ratios for these XPS peaks are summarized in Appendix A. Type II exhibits a ratio of 1:1.7 for the (Mo + W):S elements that make up MoS_2_ and WS_2_. These XPS quantitative analysis results indicate a sulfur deficiency occurred during the synthesis process of the MoS_2_/WS_2_ hybrid films.

For the investigation of the fine structures of MoS_2_ and WS_2_ within the Type I, II and III samples, we conducted focused ion beam (FIB) processing on the thin sections of the samples and observed the cross sections using high-resolution TEM (Figure 4).

Figure 4 presents the cross-sectional views of the Type I, II and III samples, showing energy-dispersive X-ray spectroscopy (EDS) mapping images of Mo (in blue) and W (in orange) obtained through scanning transmission electron microscopy (STEM) and high-resolution transmission electron microscopy (HRTEM). For Type I, the EDS mapping showed a mixed distribution of Mo and W, indicating that the film had a Mo_x_W_1−x_S_2_ alloy structure. The cross-sectional HRTEM image of Type I exhibits ambiguous shading, making it difficult to distinguish the boundary between the MoS_2_ and WS_2_ layers. In contrast, for Type II, the EDS map illustrates the vertical separation of Mo and W, whereas the high-resolution TEM image reveals distinct layer-by-layer structures between the WS_2_ and MoS_2_. In the case of Type III, the HRTEM image and EDS mapping confirmed the formation of MoS_2_ with a droplet-like dot structure, with a size of 20–30 nm, on the WS_2_ film. This phenomenon resulted from the poor wetting properties of the MoS_2_ on the WS_2_ surface in high-temperature and low-pressure environments. The comparison of WS_2_ and MoS_2_ revealed a darker appearance for the WS_2_ region, attributed to the difference in atomic numbers between Mo and W. Each film consisted of 4–5 layers. The EDS mapping results and cross-sectional HRTEM images indicated that the synthesized three types of MoS_2_/WS_2_ hybrid films had alloy-, vdWH-, and dot-like fine structures. 

To accurately analyze the structural differences observed through TEM and their correlation with the Raman spectroscopic characteristics, we conducted a detailed analysis by fitting the position distribution and gap of the E_2g_ and A_1g_ modes from the mapping data in Figure 2a. The results of this analysis are summarized in Table 1. The data used for the analysis in Figure 5 and Table 1 were exclusively extracted from the MoS_2_/WS_2_ hybrid film areas within the hexagonal grid pattern regions, excluding the areas outside the hexagonal grid pattern.

Figure 5a illustrates the distribution of the A_1g_ peak positions of the MoS_2_ and WS_2_ in the Raman spectra represented by histograms. The colors correspond to Type I (red), Type II (blue), and Type III (magenta), with the green and gray lines indicating MoS_2_ and WS_2_, respectively. Regarding the MoS_2_ A_1g_ mode at 407.6 cm^−1^, Type II exhibited a red shift of 0.5 cm^−1^, while Types I and III showed blue shifts of 2.8 cm^−1^. For the WS_2_, referencing the A_1g_ mode at 418.9 cm^−1^, Types I and II displayed red shifts of 1.4 and 0.2 cm^−1^, respectively, whereas Type III exhibited a 0.2 cm^−1^ blue shift. Figure 5b demonstrates the distribution of the E_2g_ peak positions for MoS_2_ and WS_2_ in the histograms using the same color scheme as in Figure 5a. For the MoS_2_ E_2g_ mode at 383.6 cm^−1^, Types I, II, and III showcased red shifts of 4.1, 1.5, and 2.6 cm^−1^, respectively. Regarding the WS_2_ E_2g_ mode at 357.2 cm^−1^, Types I, II and III displayed red shifts of 0.9, 2.4, and 0.8 cm^−1^, respectively. Shifts observed in the E_2g_ and A_1g_ modes of MoS_2_ and WS_2_ in the Raman spectra are typically attributed to doping or strain. However, in the MoS_2_/WS_2_ hybrid film analyzed in this study, there were no opportunities or processes for doping during the CVD synthesis. Considering the distinct microstructures observed via TEM (as described in Figure 4), it is likely that the shifts in the E_2g_ and A_1g_ modes in the Raman spectra are attributed to the strain exerted between MoS_2_ and WS_2_ at the interface. Examining the A_1g_ mode of the MoS_2_, the peak of Type I indicates a mixture of MoS_2_ and WS_2_ (as in Figure 4a), resulting in a blue shift of 2.8 cm^−1^, while Type III, characterized by MoS_2_ dot structures, experiences a vertical strain, causing a 2.8 cm^−1^ blue shift. Conversely, Type II possesses a clear vdWH structure at the MoS_2_–WS_2_ boundary, experiencing minimal strain, resulting in only a subtle 0.5 cm^−1^ red shift. Regarding the WS_2_ A_1g_ mode, Types II and III displayed minor shifts, indicating the presence of a nearly negligible strain. Type II exhibits a horizontally uniform interface in the vdWH structure, whereas Type III creates local areas for the MoS_2_ dots, contributing to the interface. In contrast, a noticeable 1.4 cm^−1^ red shift was observed for Type I, owing to vertical strain induced by the alloy microstructure. Additionally, the E_2g_ modes of MoS_2_ and WS_2_ displayed red shifts in Types I, II and III. The horizontally acting strain in all the samples aligns well with the structural characteristics. Notably, the standard deviation of the red shift in Type I was smaller than that of the other samples, implying a uniformly distributed horizontal strain attributed to vdWHs. Figure 5c shows the correlation plot of MoS_2′_s E_2g_ and A_1g_ modes [13,21]. Types I and III exhibited scattered features at strain levels between 1.0% and 0.75%, whereas Type II showed a consistent distribution at 0.5% strain. These analytical findings indicate the capability to differentiate the microstructure of TMD hybrid films by analyzing the strain-induced behavior in the E_2g_ and A_1g_ modes of the Raman spectra.

## 4. Conclusions

In this study, we synthesized MoS_2_ and WS_2_ individual single TMD films and MoS_2_/WS_2_ hybrid films, namely, Type I (Mo_x_W_1−x_S_2_ alloy), Type II (MoS_2_/WS_2_ vdWH), and Type III (MoS_2_ dots/WS_2_), via CVD using metal oxide (WO_3_ and MoO_3_) films and dilute H_2_S gas as precursors. The synthesized MoS_2_/WS_2_ hybrid films were characterized via OM and Raman mapping to confirm variations in their structural composition. Microstructural analysis for structure identification was conducted through TEM with EDS and cross-sectional imaging. Furthermore, Raman mapping facilitated a detailed comparison and analysis of the phonon behavior induced by strain at the MoS_2_ and WS_2_ interfaces. Based on their structural characteristics, distinct variations in the A_1g_ and E_2g_ modes were observed in the MoS_2_/WS_2_ hybrid films compared to those in the individual MoS_2_ and WS_2_ TMD films. Type II showed minimal changes in MoS_2_ A_1g_ mode, while Type I and III exhibited a ~2.8 cm^−1^ blue shift. In contrast, the WS_2_ A_1g_ mode in Type I showed a 1.4 cm^−1^ red shift, whereas Types II and III showed negligible or slight changes in peak positions. The alignment observed between strain-induced changes in the A_1g_ mode and structural features of MoS_2_/WS_2_ hybrid films underscores the application scope of Raman spectroscopy in discerning subtle differences in the microstructures of the Type I, II and III films. We expect that our results will promote the elucidation and identification of the structural characteristics of diverse hybrid TMD materials.

## Figures and Tables

**Figure 1 nanomaterials-14-00248-f001:**
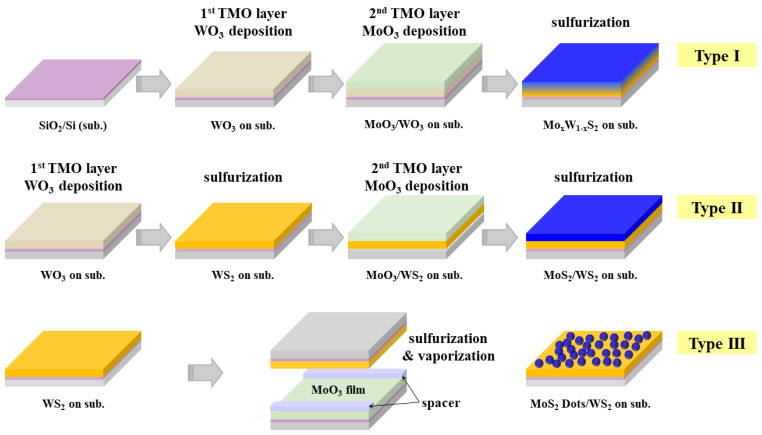
Schematic of TMD synthesis and structural diagrams of Types I, II and III.

**Figure 2 nanomaterials-14-00248-f002:**
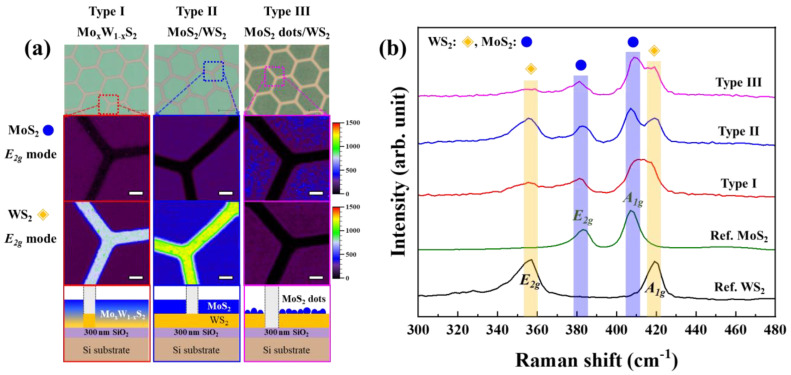
(**a**) Intensity mapping of the E_2g_ mode at 383 cm^−1^ (MoS_2_ E_2g_ mode) and 355 cm^−1^ (WS_2_ E_2g_ mode) for each structure of the MoS_2_/WS_2_ hybrid thin film. (**b**) Representative Raman spectra of each sample.

**Figure 3 nanomaterials-14-00248-f003:**
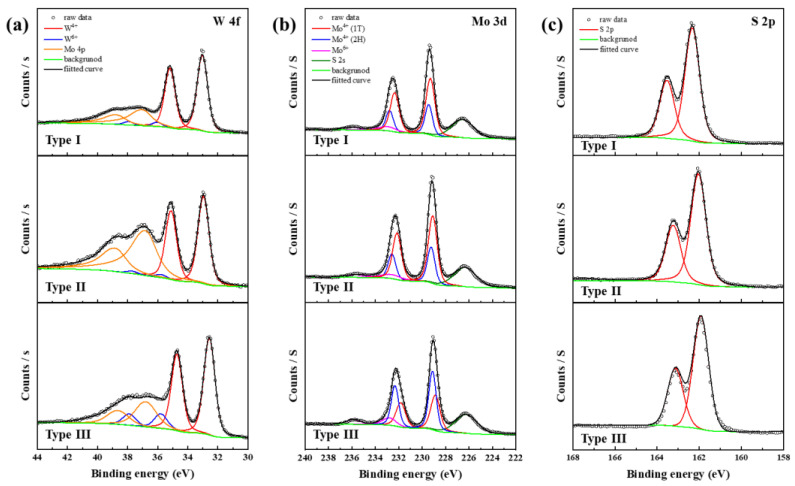
(**a**) W4f, (**b**) Mo3d, (**c**) S2p XPS spectra of the MoS_2_/WS_2_ hybrid films.

**Figure 4 nanomaterials-14-00248-f004:**
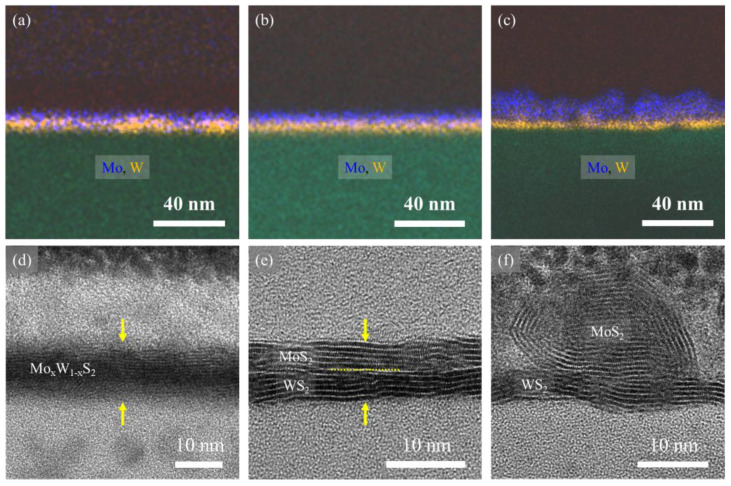
(**a**–**c**) STEM-EDS images and (**d**–**f**) cross-sectional TEM images of Types I, II and III. In the EDS map, blue corresponds to Mo, and orange corresponds to W elements.

**Figure 5 nanomaterials-14-00248-f005:**
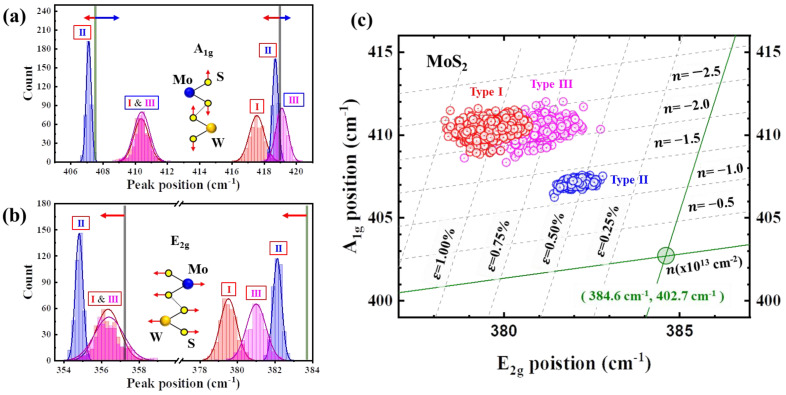
Histogram of the (**a**) A_1g_ and (**b**) E_2g_ peak positions for MoS_2_ and WS_2_ in Types I, II and III. (**c**) Correlative plot of the A_1g_ and E_2g_ peak positions for MoS_2_ to evaluate biaxial strain and charge doping distributions in Type I, II and III films. Red, blue and magenta colors indicate Type I (Mo_x_W_1−x_S_2_ alloy), Type II (MoS_2_/WS_2_ vdWH), and Type III (MoS_2_ Dots/WS_2_) films, respectively.

**Table 1 nanomaterials-14-00248-t001:** Positions of the E_2g_ and A_1g_ peaks of MoS_2_ and WS_2_ in Type I, II and III films.

		E_2g_ (cm^−1^)	A_1g_ (cm^−1^)	Gap (cm^−1^)	ɛ (%)	n (10^13^ cm^−2^)
**Type I** Mo_x_W_1−x_S_2_ alloy	MoS_2_	379.5 ± 0.4	410.4 ± 0.5	30.9	1.00	−2.1
	WS_2_	356.3 ± 0.6	417.5 ± 0.5	61.2		
**Type II** MoS_2_/WS_2_ vdWH	MoS_2_	382.1 ± 0.2	407.1 ± 0.2	25.0	0.50	−1.2
	WS_2_	354.8 ± 0.2	418.7 ± 0.2	63.9		
**Type III** MoS_2_ dots/WS_2_	MoS_2_	381.0 ± 0.5	410.4 ± 0.5	29.4	0.75	−2.1
	WS_2_	356.4 ± 0.8	419.1 ± 0.4	62.7		
	MoS_2_	383.6	407.6	24.0	0.75	−2.1
	WS_2_	357.2	418.9	61.7		

## Data Availability

The data presented in this study are available on request from the corresponding author.

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
