# Peer review of "Tailored Synthesis of Heterogenous 2D TMDs and Their Spectroscopic Characterization"

_nanomaterials, 2024, doi:10.3390/nano14030248_

Round 1

Reviewer 1 Report

Comments and Suggestions for Authors

In this manuscript, the authors synthesized heterogeneous TMDs films by using metal oxide and H2S as precursors via CVD. The Raman spectroscopy, HRTEM and EDS were used to characterize the as-synthesize films. The results are interesting and might be of interest for the researchers in the field of 2D materials. I recommend acceptance after the following concerns are well addressed.

1. How does the thickness of metal oxide influence the thickness of TMDs?

2. The authors claimed monolayer WS2 and MoS2 films were synthesized. However, no evidence was provided to confirm it.

3. As shown in Figure 3, the interlayer distance of Type I alloy is much smaller than that in type II and III. How to explain it?

4. Please double check the laser wavelength of Raman spectrometer. As shown in Figure 2b, the E2g peak has similar intensity to the A1g peak for WS2. As reported in literature (see Scientific Reports volume 3, Article number: 1755 (2013)), the E2g peak should be much stronger than the A1g peak if 514 nm laser is used.

Comments on the Quality of English Language

No comments.

Reviewer 2 Report

Comments and Suggestions for Authors

This is a really interesting paper correlating the Raman spectra of MoS2/WS2 stacks grown in different ways.

The 3 different types are very interesting, and I really enjoyed the way Fig 4 is displayed.

I'd like to know what the overall composition of the different materials is (from the EDX data or ICP), to see if that has influenced the spectroscopic data. 

Reviewer 3 Report

Comments and Suggestions for Authors

Authors reported the chemical vapor deposition (CVD) growth of a MoS2/WS2 hybrid system and characterized it using Raman scattering. Depending on the growth conditions, three kinds of hybrid structures were synthesized, and the strain and identification of each structure were determined from peak analyses of A1g and E2g modes. Mapping the MoS2 peak with strain and carrier concentration revealed clear differences between the series. The results appear to be useful for non-destructive characterizations of the MoS2/WS2 hybrid system. Before publishing the manuscripts, the authors should address the following questions and comments:

  1. 1. Details of analyses, such as the estimation of peak position, strain, and carrier concentration, should be summarized in the Materials and Methods section with cited references. If the present results can be used for the characterization of the MoS2/WS2 hybrid system, it will be essential information for readers.

  2.  
  3. 2. Regarding the mapping of A1g and E2g of MoS2 (Fig. 4c), types I and III have almost the same carrier concentration. Additionally, the energy-dispersive X-ray spectroscopy (EDS) mapping of III shows a mixing of yellow and blue colors, indicating that the type III MoS2 droplet is Mo1-xWxS2. The authors should summarize the compositions of the top and bottom regions of the I, II, and III hybrid systems from analyses of EDS spectra.

Round 2

Reviewer 1 Report

Comments and Suggestions for Authors

I have no question for this version and recommend acceptance in its current form.

Comments on the Quality of English Language

Minor editing of English language required.